# The Relationship between Mastitis and Antimicrobial Peptide S100A7 Expression in Dairy Goats

**DOI:** 10.3390/vetsci10110653

**Published:** 2023-11-14

**Authors:** Yutong Yan, Kunyuan Zhu, Haokun Liu, Mingzhen Fan, Xiaoe Zhao, Menghao Pan, Baohua Ma, Qiang Wei

**Affiliations:** 1Key Laboratory of Animal Biotechnology of the Ministry of Agriculture, Northwest A&F University, Xianyang 712100, China; yan-yutong@nwafu.edu.cn (Y.Y.); kunyuan_zhu@163.com (K.Z.); lhkxn@nwafu.edu.cn (H.L.); 18437963227@163.com (M.F.); zhaoxe@nwafu.edu.cn (X.Z.); panmenghao@nwafu.edu.cn (M.P.); 2College of Veterinary Medicine, Northwest A&F University, Yangling 712100, China

**Keywords:** S100A7, somatic cell count, dairy goat, mastitis

## Abstract

**Simple Summary:**

Antimicrobial peptides, particularly S100A7, have beneficial effects on host defenses. The study aims to evaluate the relationship between mastitis and antimicrobial peptide S100A7 expression in dairy goats. The somatic cell count (SCC), milk S100A7 concentration, S100A7 immunohistochemistry and expression of udder were measured. Results showed that the level of S100A7 expression was significantly upregulated in the mammary gland of mastitis dairy goats than in healthy dairy goats. S100A7 was expressed weakly in the alveolus of a healthy goat mammary gland, while densely in the collapsed alveolus of a mastitis goat mammary gland. S100A7 concentration was significantly upregulated in subclinical mastitis goats and had no significant difference in clinical mastitis goats compared to healthy dairy goats. The S100A7 concentration in milk has a limited relationship with SCC or mastitis.

**Abstract:**

S100A7 is an inflammation-related protein and plays an essential role in host defenses, yet there is little research about the relationship between mastitis and S100A7 expression in dairy goats. Here, according to the clinical diagnosis of udders, SCC, and bacteriological culture (BC) of milk, 84 dairy goats were grouped into healthy goats (*n* = 25), subclinical mastitis goats (*n* = 36), and clinical mastitis goats (*n* = 23). The S100A7 concentration in subclinical mastitis goats was significantly upregulated than in healthy dairy goats (*p* = 0.0056) and had a limited change with clinical mastitis dairy goats (*p* = 0.8222). The relationship between log_10_ SCC and S100A7 concentration in milk was positive and R = 0.05249; the regression equation was Y = 0.1446 × X + 12.54. According to the three groups, the log_10_ SCC and S100A7 were analyzed using the receiver operating characteristics (ROC) curve; in subclinical mastitis goats, the area under the ROC curve (AUC) of log_10_ SCC was 0.9222 and *p* < 0.0001, and the AUC of S100A7 concentration was 0.7317 and *p* = 0.0022, respectively; in clinical mastitis goats, the AUC of log_10_ SCC was 0.9678 and *p* < 0.0001, and the AUC of S100A7 concentration was 0.5487 and *p* = 0.5634, respectively. In healthy goats, S100A7 was expressed weakly in the alveolus of the mammary gland of healthy goats while expressed densely in the collapsed alveolus of mastitis goats. Moreover, *S100A7* expression increased significantly in mastitis goats than in healthy dairy goats. In this research, results showed the effects of mastitis on the S100A7 expression in the mammary gland and S100A7 concentration in milk and the limited relationship between SCC and mastitis, which provided a new insight into S100A7’s role in the host defenses of dairy goats.

## 1. Introduction

Mastitis is the most significant health problem in animal farming, usually caused by microbial infection, impedes curd formation, and results in the reduction of milk yield [1,2] and quality of milk [3]. At present, the use of antibiotics makes milk unable to meet the requirements of human no-antibiotic milk and has led to the emergence of resistant microbes [4,5]. Mastitis is still a common disease in farming and causes huge economic losses all over the world [3,6]. Based on the diagnosis of milk and udder, mastitis can be subdivided into subclinical and clinical mastitis [7], in which visual alterations in milk and udder can be detected in clinical mastitis [7], such as the presence of udder or quarter swelling, redness, heat, hardness, pain, and visible alterations of the milk. SCC, a modern management strategy and udder health monitoring plan, is also a parameter for monitoring mastitis to decrease its impact on cows [8,9]. However, in small ruminants, especially in goats, numerous non-infectious factors influence SCC [10], such as seasons, lactation, and parity, and also have a significant variation of SCC among individuals and farms [11]. Moreover, the style of holocrine secretion in goat milk also cannot be ignored, which causes the physiological shedding of cytoplasmic particles to increase the threshold value of SCC in healthy goats [12,13]. The reports still recommend that the SCC of <5.0 × 10^5^ or >1.0 × 10^6^ cells/mL should reliably indicate the absence or presence of subclinical mastitis, respectively [14,15,16]. Of course, the BC of milk is still the best way to precisely detect mastitis in the herd [17], yet it is affected by cost and sensitivity issues and the applicability of routine screening.

The antibacterial factor belongs to the innate immune system and can prevent bacteria from destroying normal conditions. The antimicrobial peptide (AMP) can target a broad spectrum of microbes, which is gaining more and more attention [18,19]. More specifically, S100A7, also known as psoriasin, is an EF-hand type calcium-binding protein [20] isolated from the abnormally proliferated keratinocytes of psoriasin patients in 1991 [21]. S100A7 is localized in the epithelial cells [22] and forms homodimers by non-covalent bindings [23,24] and is also found in bovine [25] and goat [26], yet the research about the effect of mastitis on its expression in small ruminants remains unclear. AMPs are potential protein markers of mastitis monitoring, such as cathelicidins [27,28] and the lingual antimicrobial peptide [29], and detect milk concentration to monitor mastitis [28,30]. Moreover, specific subtypes of inflammatory cells are attracted by S100A7 [31], which is considered an inflammation-related protein with the effect of chemokines [32]. Interestingly, these specific cells are important parts of milk somatic cells, but the relationship between S100A7 expression and the concentration of SCC is still unknown. Furthermore, S100A7 concentration is significantly increased after an intramammary infusion with lipopolysaccharide (LPS) [26]; whether the abundance of S100A7 correlates with the degree of inflammation associated with mastitis in goats is still unclear. In a word, S100A7 plays a potential role in the innate immune system of the mammary gland, yet the relationship between mastitis and S100A7 expression in dairy goats is still unknown. With these premises, studying the effects of mastitis on the S100A7 expression in the mammary gland and milk of dairy goats provides a new insight into S100A7’s role in host defenses.

## 2. Materials and Methods

### 2.1. Milk Sample and Goat Mammary Tissue Sample

The experimental protocols and procedures in the study were reviewed and approved by the Institutional Animal Care and Use Committee of the College of Veterinary Medicine, Northwest A&F University (Approval No. 2019031502). Considering that mastitis frequently develops in the early stages of the whole lactation, it occurs with a higher incidence during the first 4 weeks after parturition, and with 74–95% of cases occurring during the first 3 months [33,34,35,36], so a total of 84 Guanzhong dairy goats (range in parity from 1 to 3; body weight: 55 ± 4.8 kg) in the first three months of lactation were used in this study, selected randomly from the Hongxing Meiling Dairy Goat breeding Farm located in Fuping, Shannxi, China. All the experimental dairy goats were housed in free stalls and allowed unlimited access to feed and water. The nutrient requirement of dairy goats could be satisfied according to the Nutrient Requirements of Small Ruminants (National Research Council, 2007), including alfalfa hay, corn, wheat bran, soybean meal, wheat straw, corn silage, corn germ meal, cottonseed meal, calcium hydrophosphate, limestone, sodium carbonate, sodium chloride, and mineral and vitamin premix; the chemical composition of the diets were presented in Appendix A. The diet was provided twice a day at 7:30 and 15:30. Healthy udder is defined as a non-inflammatory condition, and the SCC is lower than 5.0 × 10^5^ cells/mL [14,15,16]; clinical mastitis was defined firstly as the presence of clinical mastitis symptoms such as the presence of udder or quarter swelling, redness, heat, hardness, pain, visible alterations of the milk, such as watery appearance, clots, flakes, or pus [37,38], while subclinical mastitis was defined without these clinical symptoms, and the bacteriological culture of milk in subclinical mastitis and clinical mastitis were positive. After disinfecting the teat of goat mammary three times with 75% alcohol and discarding the first three handfuls of milk, milk was sampled in the morning at one time; about 40 mL of milk was sampled with a sterile tube (50 mL), transported at low temperature for somatic cell staining or immediately transported with dry ice for detection of S100A7 concentration. Twelve goat mammary tissues were sampled from the Fuping slaughterhouse in ShaanXi Province, China, which were fixed in 4% paraformaldehyde to perform paraffin secretions, or immediately frozen in liquid nitrogen for RNA extraction [39].

### 2.2. Somatic Cell Count Determination

The milk somatic cell count determination was used with a direct microscopic somatic cell count (DMSCC). The SCC was counted according to the Diff–Quik Stain manual (Solarbio, G1540, Beijing, China) according to previous researches [40,41]. Briefly, a mixed sample, a pipette of 10 μL of the sample, was placed in the center of a glass slide; the tip of the pipette gently drew a circle and averagely coasted it into an area of 1 cm^2^, fixed formaldehyde for 5 min, immersed in Diff–QuickIDyeing for 5 min and immersion in Diff–QuickIIdyeing for 5 min, 95% alcohol decolored for 5 s, 100% alcohol decolored for 30 s, transparency with xylene for 1 min and sealing, and observed under a microscope. Count the number of somatic cells in 30 fields of view, then calculate the SCC according to the formula: SCC = coefficient × average number of cells per field of view (total number of cells ÷ 30), that was the number of somatic cells per milliliter of milk sample. The SCC was not normally distributed, which was analyzed in log10 to normalize the distribution [42,43].

### 2.3. Bacteriological Culture

The bacteriological culture of goat milk was performed according to the previous reports [44,45]. Firstly, milk samples were hand-mixed three times and opened in a biosafety level II cabinet. A total of 10 µL of milk was spread over an agar base supplemented with 5% sheep blood, and the plates were incubated at 37 °C. The number of colonies on the plate was counted after 24 h to 48 h and obtained the colony-forming unit (CFU). After the first 24 to 48 h of incubation, if no growth was visible on the plates, the samples were reincubated (37 °C) and rechecked at 24 h intervals for up to 96 h. No viable colony was considered healthy; otherwise, it was mastitis.

### 2.4. Immunohistochemistry

The immunohistochemistry of goat mammary tissues was performed according to the previous report [46]. The goat mammary tissues were fixed in 4% paraformaldehyde (Solarbio, P1110, Beijing, China) and kept at 4 °C. Tissues were dehydrated with different concentrations of ethanol and embedded in paraffin overnight, sections were continuously sliced with a thickness of 3 µm, and one of every five sections was selected and used for immunohistochemistry. Paraffin sections were deparaffinized with xylene and hydrated in graded ethanol series before staining using the streptavidin–peroxidase method. Antigens were retrieved by boiling for 20 min in a citrate antigen retrieval solution (Solarbio, C1031, Beijing, China). Endogenous peroxidase was blocked by incubation in 3% hydrogen peroxide. The primary antibody of anti-S100A7 (1:100, Bioss, bs-6238R, Beijing, China) was incubated overnight at 4 °C, and the second antibody was incubated at 37 °C for 1 h. After incubation, sections were lightly counterstained with hematoxylin, dehydrated, and coverslipped, and sections were viewed and captured under a microscope.

### 2.5. Enzyme-Linked Immunosorbent Assay (ELISA)

The S100A7 concentration in goat milk was assessed using ELISA (MEIMIAN, MM-7510501, Yancheng, China) according to its instruction; the intra- and inter-assay CV’s of the ELISA were less than 10% and 15%, respectively; the limitation was 1.25–40 μg/mL. The absorbance was detected at 450 nm using the microplate reader (Tecan, Group Ltd., Mennedorf, Switzerland). Briefly, first mix the samples, set blank wells, standard wells, and test the sample wells, and add 50 μL of standard to the standard well; add prepared samples to the sample well; add 100 μL of a HRP-conjugate reagent to each well, except the blank well, react for 60 min at 37 °C; wash five times with a washing buffer, add 50 μL of chromogen solution A and B to each well, develop color at 37 °C for 10 min, add 50 μL of a stop solution to stop the reaction, then read the OD value within 15 min, and calculate the concentration.

### 2.6. Total RNA Isolation and cDNA Synthesis

Total RNA isolation was obtained using a MiniBEST Universal RNA Extraction Kit (TaKaRa, 9767, Dalian, China), and the cDNA was synthesized using a PrimeScript RT Master Mix reverse transcription kit (TaKaRa, RR036, Dalian, China), according to the manufacturer’s instructions. The total RNA was 0.2 μg, the 5× PrimeScript RT Master Mix was 2 μL, and RNase Free ddH_2_0 was up to 10 μL. The RT-PCR parameters were as follows: 37 °C for 15 min and 85 °C for 5 s. 

### 2.7. q-PCR

S100A7 goat-specific primers 5′-CCAGCAAGGACAGGAACTCA-3′ for forward and 5′-GCAGCTGCTGAAGGAGAACT-3′ for reverse. The expression of GAPDH was analyzed as an internal control; its primers were 5′-TGCCCGTTCGACAGATAGC-3′ for forward and 5′-ACGATGTCCACTTTGCCAGTA-3′ for reverse. The amplified products were separated and analyzed via electrophoresis on a 3% agarose gel (Solarbio, a8350, Beijing, China). The quantitative RT-PCR reactions were performed using the SYBR Green Premix *Pro Taq* HS qPCR Kit (Accurate Biotechnology Co., Ltd., AG11718, Changsha, China), and data collection were performed on the QuantStudio 6 Flex machine (Invitrogen Corporation, Waltham, MA, USA). cDNA 0.2 μg, 2 × SYBR Green *Pro Taq* HS 10 μL, Forward Primer and Reverse Primer (10 μM) 0.2 μL, RNase Free ddH_2_0 up to 20 μL. The quantitative RT-PCR parameters were as follows: 95 °C for 30 s, followed by 40 cycles each of 95 °C for 5 s, and 60 °C for 30 s. The level of mRNA quantification was estimated using the 2^−ΔΔct^ method. Relative gene expression was obtained by normalizing with the GAPDH expression, calculating differences in mRNA expression as fold changes relative to expression in the control group.

### 2.8. Statistics

Results were presented as the mean ± S.E.M. without less than three replicates for each experimental condition. The paired groups were applied to compare with unpaired Student’s *t*-tests after confirming the normal distribution; the multiple groups were applied with a one-way analysis of variance (ANOVA) followed by Tukey’s multiple comparison test with GraphPad Prism software (version 8.0). A value of *p* > 0.05 was defined as no significant difference and represented with “ns”, a value of *p* < 0.05 was defined as a significant difference and represented with “*”, and a value of *p* < 0.01 was defined as an extremely significant difference and represented with “**”.

## 3. Results

### 3.1. The Difference of S100A7 Concentration in Subclinical and Clinical Mastitis Goats

The represented images of somatic cells stained with the Diff–Quik stain were presented in Appendix A. Figure 1 illustrated the difference in S100A7 concentration from 84 dairy goat milk samples. Compared with healthy goats, S100A7 concentration was significantly upregulated in subclinical mastitis goats (Figure 1, *p* < 0.01), while it had no significant difference in clinical mastitis goats; S100A7 concentration in subclinical mastitis samples was significantly higher than clinical mastitis goats (Figure 1, *p* < 0.05).

### 3.2. Test Characteristics of SCC and S100A7 Levels in Subclinical and Clinical Mastitis Goats

For assessing the respective test characteristics of sensitivity and specificity [47,48], the ROC analysis was generated for the SCC and S100A7 concentration from the collected samples according to the group, and the AUC was calculated. Results were shown in Figure 2. In subclinical mastitis goats, the AUC of log_10_ SCC was 0.9222, *p* < 0.0001, and the AUC of S100A7 concentration was 0.7317, *p* = 0.0022; in clinical mastitis goats, the AUC of log_10_ SCC was 0.9678, *p* < 0.0001, and the AUC of S100A7 concentration was 0.5487, *p* = 0.5634.

### 3.3. Relationship between the SCC and S100A7 Concentration in Dairy Goat

In order to measure the relationship between the S100A7 concentration in milk and the SCC, the log_10_ SCC and S100A7 concentration of 84 dairy goats were used. The relationship between the log_10_ SCC and the S100A7 concentration in milk was positive, R = 0.05249, and the regression equation was Y = 0.1446 × X + 12.54 (Figure 3, *p* > 0.05).

### 3.4. The Immunohistochemistry and Expression of S100A7 in Healthy and Clinical Mastitis Dairy Goat Gland

Immunohistochemistry analysis showed that S100A7 was expressed weakly in the alveolus of a healthy goat mammary gland but not in the connective tissue (Figure 4A,B). However, the S100A7 immunoreactivity could be observed densely in the collapsed alveolus of the clinical mastitis goat mammary gland (Figure 4E,F). Compared with healthy goat mammary glands, the S100A7 expression was upregulated significantly in clinical mastitis goat mammary glands (Figure 4I, *p* < 0.01).

## 4. Discussion

Mastitis is the most serious problem of mammary, which causes huge economic losses in the milk industry worldwide [3,6]. At present, antibiotic is the popular method of prevention and treatment for mastitis, but it affects the milk quality and causes resistance [4,5], so an alternative strategy that used to prevent and treat mastitis is necessary for the dairy industry’s development. The AMPs are receiving more and more attention for their broad-spectrum antibacterial activity and almost no resistance [18,19], becoming valuable substitutes for preventing and treating mastitis, which can satisfy the requirement of antibiotic-free milk for humans. The SCC is considered an important parameter of milk quality and udder health [49], which is used to monitor mastitis to decrease its impact. The SCC could be performed in Fossomatic [41], yet traditional detection methods of mastitis, such as bacteriological culture, DMSCC, and the California mastitis test (CMT) [50], are still the main methods for diagnosing mastitis.

DMSCC is a convenient method that can be used by trained technical persons/veterinarians under field conditions and is considered a reliable method in ewe milk [41]. In this study, somatic cells were stained using the Diff stain and counted, yet the insufficiency of the subjective judgment still needs attention, which is easily influenced by the observer. Milk somatic cells are a natural component that comprise epithelial cells and leukocytes [51]. In the present study, the SCC in mastitis goat milk was increased visibly, which is similar to the previous research [51]; different cell types were observed, such as macrophages, lymphocytes, neutrophils, and red blood cells. Moreover, red blood cells were visible in mastitis dairy goat milk, which showed that the blood–milk barrier was destroyed in the mammary gland of a mastitis goat. It is recommended that the SCC of <5.0 × 10^5^ or >1.0 × 10^6^ cells/mL could reliably indicate a healthy goat or the presence of subclinical mastitis, respectively [14,15,16]. In the present study, although the bacteriological culture of goat milk was positive in subclinical or clinical mastitis goats, the SCC might be lower than 1 × 10^6^ cells/mL, even 5.0 × 10^5^ cells/mL; even though the parity of goat in each group was the same, there was some degree of fluctuation in the SCC. In a word, these results further supported that SCC was considered an indirect index in goats, which could be influenced by non-infectious factors, such as parity, seasons, and lactation, and eventually led to the limited performance of the SCC for monitoring mastitis and no reliable threshold values in small ruminants.

Various antimicrobial components play essential roles in the defense of mammary, which are synthesized and secreted into milk in the mammary glands [52]; AMPs are the important members, such as cathelicidins, lactoferrin, and defensin. AMPs are potential protein markers of mastitis monitoring, have attracted more and more attention for the relationship with SCC, and have developed ELISA to monitor mastitis [28,30]. S100A7 is a number of AMPS with a strong antimicrobial activity. In this study, the S100A7 concentration not only has a small correlation with SCC, but also the AUC was lower than the log_10_ SCC in subclinical or clinical mastitis goat, and the correlation coefficient between the SCC and S100A7 concentration was lower than other AMPs, which show the limited monitor for mastitis. Circulating substances may play vital roles in S100A7 synthesis and secretion; the previous reports have shown that butyrate enhanced the production of S100A7 in the mammary epithelial cells, and butyrate enhance S100A7 concentrations via the infusion into mammary glands via teats [53]. Because S100A7 is a calcium-binding protein [21], S100A7 concentration is expectedly related to calcium concentration. Circulating leukocytes are the major source of immune cells and defense mechanisms and are also the main component of milk somatic cells, which may be the source of S100A7 for that S100A8-positive cells are leukocytes in goat milk [54], yet the relationship is still unclear.

In this study, S100A7 was presented in the filling and completing alveolus of a healthy goat mammary gland; when pathogenic microorganisms invaded the mammary gland of a goat, S100A7 was synthesized and secreted into milk to resist invasion. Interestingly, S100A7 concentration was significantly higher in subclinical mastitis, while it was slightly upregulated in clinical mastitis goats, which may reduce the breakdown of alveolus function in clinical goats that disordered the source of S100A7. Especially in clinical mastitis goats, curds can be observed easily, and milk production would be reduced. Moreover, S100A7 secretion is released in a biphasic manner in human epidermal keratinocytes, with different characteristics in the acute phase or chronic phase [55]. In goat mammary epithelial cells, the secretion of S100A7 depends on the time and concentration of LPS treatment, reaching a peak at about 12 h and starting to decline within 48 h, rather than increasing with prolongation of treatment time, show that S100A7 is synthesized secreted in a rapid and short-time manner [46], so S100A7 may be secreted into milk in the same style as the mammary epithelial cells in dairy goat.

The individual difference may affect its reliability; the basal S100A7 concentration in healthy goat samples (8.36–15.20 μg/mL) had a low degree of centralization in this study, which is similar to the previous research [26]. Similar phenomena were presented in subclinical or clinical mastitis goats. In this work, we pay more attention to the period that mastitis is a high incidence in the first three months of lactation. The entire lactation has not been studied, so it is limited and does not represent the entire lactation stage. Types of pathogenic microorganisms are of equal importance, especially *E. coli*, which is mainly against invasion by S100A7 antibacterial activity [56,57], while it has low or no antibacterial activity for other pathogenic bacteria, such as *Staphylococcus aureus* [25]; yet the germline of *E. coli* doesn’t play a decisive role in the severity of its induced mastitis, differences in the main strains causing subclinical and clinical mastitis also lead to differences of S100A7 abundance, and the relationship between S100A7 abundance and the different strains remains unclear. The teat epithelium plays a vital role in host defense, which is considered the main place for the synthesis and secretion of S100A7 in bovine [58] and goat [26], so the source of keratinocytes is also an important role in studying S100A7 expression. S100A7 concentration in milk increase upon frequent teat stimulation both with or without milk removal [59]. Moreover, inflammatory cytokines influence its synthesis and secretion, such as IL-1β, which can induce S100A7 production [60]. In a word, S100A7 plays an important role in the innate immune defense of the goat mammary gland, yet there are many factors that affect its synthesis and secretion, which is necessary for further study.

## 5. Conclusions

The level of S100A7 expression was upregulated in the mammary gland of mastitis dairy goats than in healthy dairy goats. S100A7 was expressed weakly in the alveolus of a healthy goat mammary gland, while densely in the collapsed alveolus of a mastitis goat mammary gland. S100A7 concentration was significantly upregulated in subclinical mastitis goats and had no significant difference in clinical mastitis goats compared to healthy dairy goats. The S100A7 concentration in milk has a limited relationship with SCC or mastitis, which show the limited monitoring for mastitis in dairy goat.

## Figures and Tables

**Figure 1 vetsci-10-00653-f001:**
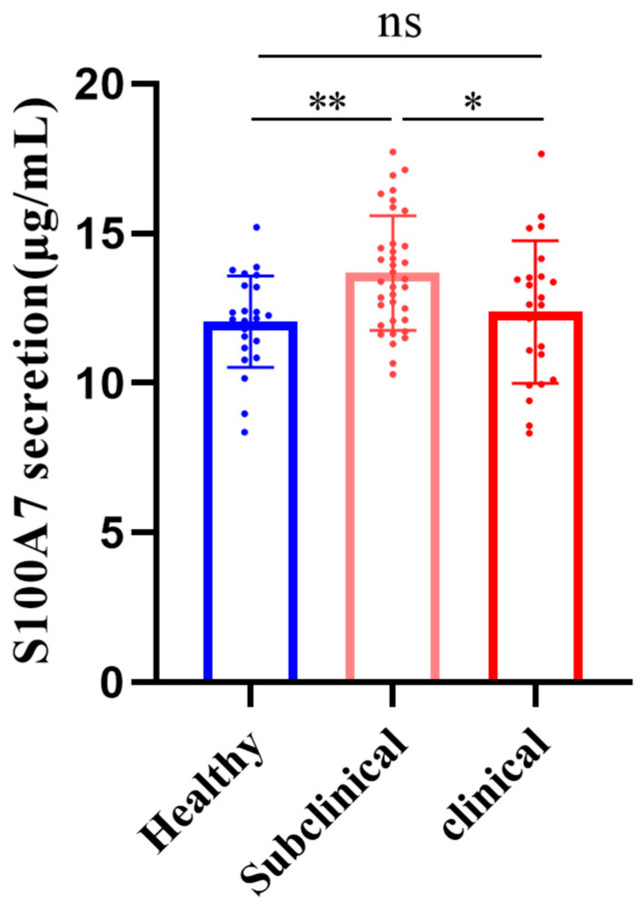
The difference in S100A7 concentration from 84 samples. Healthy goats (Healthy, *n* = 25), subclinical mastitis goats (Subclinical, *n* = 36), and clinical mastitis goats (Clinical, *n* = 23); ns: *p* > 0.05; *: *p* < 0.05; **: *p* < 0.01.

**Figure 2 vetsci-10-00653-f002:**
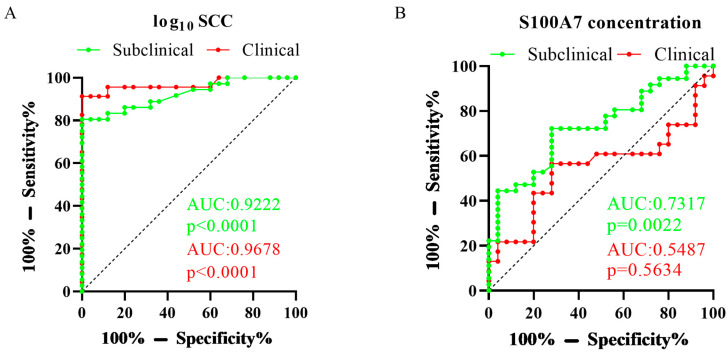
The ROC analysis of log_10_ SCC and S100A7 levels based on the three groups. (**A**) The ROC analysis of log_10_ SCC in subclinical mastitis goats and clinical mastitis goats. (**B**) The ROC analysis of S100A7 concentration in subclinical mastitis goats and clinical mastitis goats. SCC: somatic cells count; AUC: the area under the curve (AUC); *p*: *p*-value.

**Figure 3 vetsci-10-00653-f003:**
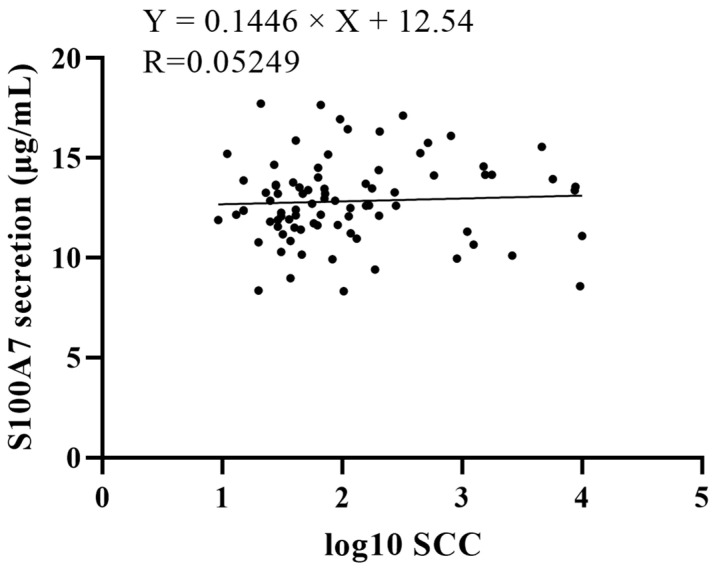
The regression equation between S100A7 concentration and log_10_ SCC from 84 samples. SCC: somatic cell count; R: Pearson r.

**Figure 4 vetsci-10-00653-f004:**
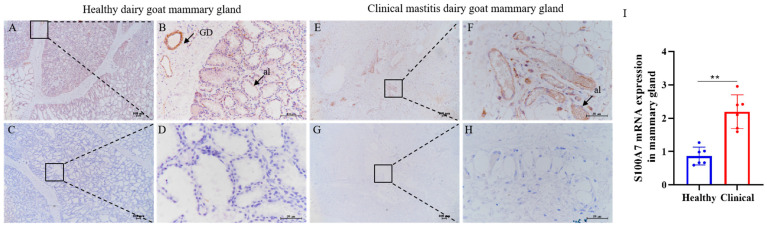
Immunohistochemistry and expression of S100A7 in healthy dairy goat and clinical mastitis dairy goat mammary gland. (**A**,**B**) The represented images of healthy dairy goat (*n* = 6) mammary gland S100A7 antibody staining; (**C**,**D**) the represented images of healthy dairy goat mammary gland negative control; (**E**,**F**) the represented images of clinical mastitis dairy goat (*n* = 6) mammary gland S100A7 antibody staining; (**G**,**H**) the represented images of clinical mastitis dairy goat mammary gland negative control; (**I**) the analysis of S100A7 mRNA expression level in alveolus between healthy (Healthy, *n* = 6) and clinical mastitis mammary gland (Clinical, *n* = 6); al = alveolus; GD = gland duct. **: *p* < 0.01. (**A**,**C**,**E**,**G**): scale bar = 100 μm; (**B**,**D**,**F**,**H**): scale bar = 50 μm.

## Data Availability

All data generated or analyzed during this study are included in this published article and its Appendix A.

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
