# Peer review of "The Relationship between Mastitis and Antimicrobial Peptide S100A7 Expression in Dairy Goats"

_vetsci, 2023, doi:10.3390/vetsci10110653_

Round 1
Reviewer 1 Report
Comments and Suggestions for Authors
Although the subject is not innovative, the article provides incremental information regarding the relationship between the concentration of the S100A7 protein in milk with the somatic cell count and mastitis in goats, in a way that could contribute to the continuity of research on the subject.
Author Response
Comment: Although the subject is not innovative, the article provides incremental information regarding the relationship between the concentration of the S100A7 protein in milk with the somatic cell count and mastitis in goats, in a way that could contribute to the continuity of research on the subject.
RE: Thanks for reviewer’s objective and positive comments of the subject. In this study, we studied the relationship between mastitis and antimicrobial peptide S100A7 from somatic cell count, milk S100A7 concentration, S100A7 immunohistochemistry and expression of udder in dairy goat. Results showed that the level of S100A7 expression was significantly upregulated in mammary gland of mastitis dairy goats than healthy dairy goats; S100A7 was present weakly in the alveolus of healthy goat mammary gland while densely in collapsed alveolus of mastitis goat mammary gland; S100A7 concentration was significantly upregulated in subclinical mastitis goats and had no significant difference in clinical mastitis goats compare to healthy dairy goats, and the S100A7 concentration in milk has a limited relationship with SCC or mastitis. In a word, we explored the relationship between the mastitis and S100A7 expression in dairy goats, providing a new insights of S100A7 role in host defense.
Reviewer 2 Report
Comments and Suggestions for Authors
Dear authors,
After carefully reading your manuscript, I have a few remarks and suggestions, for the manuscript to move forward:
- Line 26: include the p value;
- Line 35: SCC has limited relationship with mastitis? That does not sound well to me, since SCC is an accepted proxy world-wide to study udder health in dairy species;
- Line 55: introduce BC meaning, since you mention it here first;
Line 76-78: Rewrite the aim to sound like an aim, and not a justification, and also, write the mammary gland, not just simply 'gland';
Lines 96-97: The slaughter goats were among the 84 animals studied? If so, how many 'healthy', and how many with the two types of mastitis (sub-clinical and clinical)? Please clarify this aspect;
Were the study herd goats balanced for the same breed, parity, lactation length and production levels? Which breed were the goats?
At what intervals the milk samples were taken? Please give more details on sampling, this is a must, if others are to replicate your study;
Line 197-198: There are a wide variety of equipment for measuring SCCs, please delete this sentence;
Lines 207-2013: Please rewrite this sentence, is very confusing to the reader, make it clearer;
References 10 and 14 that the authors use to set the threshold and explain their approaches and results, are previous studies on sheep milk, thus have limited value for comparison reasons with the current study, please choose papers on goat milk and SCCs.
Overall, the manuscript has some merits, and it is worth considering for publication in this journal, however, some amendments need to be done before acceptance.
Author Response
Comment: After carefully reading your manuscript, I have a few remarks and suggestions, for the manuscript to move forward:
RE: Thanks for reviewer to spend time reading the manuscript carefully, and offered helpful and constructive suggestions, we have seriously thought about the suggestions and provided our responses in a point-by-point manner.
Question 1: Line 26: include the p value;
RE: Thanks for reviewer’s advice. “S100A7 concentration in subclinical mastitis goats was significantly upregulated than healthy dairy goats, and had a limited change with clinical mastitis dairy goats.” has revised into “S100A7 concentration in subclinical mastitis goats was significantly upregulated than healthy dairy goats (p=0.0056), and had a limited change with clinical mastitis dairy goats (p=0.8222).”
Question 2: - Line 35: SCC has limited relationship with mastitis? That does not sound well to me, since SCC is an accepted proxy world-wide to study udder health in dairy species;
RE: Thanks for reviewer’s reminding of the limit of SCC in small ruminants. Currently, the SCC is a valuable indicator used in the diagnosis of mastitis in cows, especially in the case of subclinical mastitis, which does not evolve with visible signs of mammary gland damage, but only with changes in milk composition (1). However, there is no reliable threshold values are defined in goat. On the one hand, even though the somatic cell count was less than 1×106 or even 5×105 cells/mL, it’s possible that the bacteriological culture of milk was positive, yet these threshold values were defined obviously this is unreliable. On the other hand, somatic cell count was also affected by non-infectious factors, such as seasons, lactation, parity and others. Moreover, the SCC can increase in normal dairy goats following apocrine secretion in the absence of the infection in the udder, leading to false-positive results from the research “Direct and indirect measurement of somatic cell count as indicator of intramammary infection in dairy goats”. In a word, somatic cell count couldn’t serve as a direct indicator in goat.
Question 3: - Line 55: introduce BC meaning, since you mention it here first;.
RE: Thanks for reviewer’s reminding of the “BC”. BC refer to bacteriological culture, appeared firstly in Line 24.
Question 4: Line 76-78: Rewrite the aim to sound like an aim, and not a justification, and also, write the mammary gland, not just simply 'gland';
RE: Thanks for reviewer’s advice.
- The aim was revised into “In a word, S100A7 plays a potential role in innate immune system of mammary gland, yet the relationship between mastitis and S100A7 expression in dairy goat is still unknown. With these premises, studying the effects of mastitis on the S100A7 expression in gland and milk of dairy goat, which provide a new insight of S100A7 role in host defenses.”.
- “gland”was revised into “mammary gland”
Question 5: Lines 96-97: The slaughter goats were among the 84 animals studied? If so, how many 'healthy', and how many with the two types of mastitis (subclinical and clinical)? Please clarify this aspect;Were the study herd goats balanced for the same breed, parity, lactation length and production levels? Which breed were the goats?At what intervals the milk samples were taken? Please give more details on sampling, this is a must, if others are to replicate your study;
RE: Thanks for reviewer’s advice and reminding.
- The slaughter goats weren’t among the 84 animals studies. 12 goat mammary tissues were collected from slaughterhouse and used to study the expression of S100A7 in mammary gland, milk of 84 goats were samples to detect S100A7 concentration.
- The goats were Guanzhong dairy goat, a total of 84 dairy goats (range in parity from 1 to 3; body weight: 55 ±8 kg )milk in the first three month of lactation were used in this study, were randomly selected under the same environmental and managerial conditions. The nutrient requirement of dairy goat could be satisfied according to the Nutrient Requirements of Small Ruminants (National Research Council, 2007), including alfalfa hay, corn, wheat bran, soybean meal, wheat straw, corn silage, corn germ meal, cottonseed meal, and minerals. The diet was offered twice daily at 7:30 and 15:30.
- Samples were collected in themorning at one time, about 40 mL of milk was sampled by sterile tube (50 mL) after disinfecting three times with 75% alcohol and discarding the first three handfuls milk, transportation with low temperature for somatic cell staining or immediately transported with dry ice for detection of S100A7 concentration.
These points were revised in the manuscript.
Question 6: Line 197-198: There are a wide variety of equipment for measuring SCCs, please delete this sentence;
RE: Thanks for reviewer’s advice. “ There are a wide variety of equipment for measuring SCCs” has been deleted in the manuscript.
Question 7: Lines 207-213: Please rewrite this sentence, is very confusing to the reader, make it clearer. References 10 and 14 that the authors use to set the threshold and explain their approaches and results, are previous studies on sheep milk, thus have limited value for comparison reasons with the current study, please choose papers on goat milk and SCCs..
RE: Thanks for reviewer’s advice.
- The sentence was revised into “In this study, SCC was stained by Diff-Stain and counted, yet the insufficiency of the subjective judgement still needs attention, which is easily influenced by the observer. Milk somatic cells are a natural component, which comprise epithelial cells and leukocytes. In present study, SCC in mastitis goat milk was increased visibly, which was similar to the previous research.”.
- The relative researches were cited in References 14-16 of the manuscript.
Question 8:Overall, the manuscript has some merits, and it is worth considering for publication in this journal, however, some amendments need to be done before acceptance..
RE: Thanks for reviewer’s positive comments and offered helpful and constructive suggestions. We have seriously thought about the suggestions and provided our responses in a point-by-point manner, and as a result, we feel that the manuscript has been significantly improved.
Reference
- M. J. Paape, B. Poutrel, A. Contreras, J. C. Marco, A. V. Capuco, Milk Somatic Cells and Lactation in Small Ruminants. Journal of Dairy Science84, E237-E244 (2001).
Reviewer 3 Report
Comments and Suggestions for Authors
Mastitis is one of the most significant health problems in dairy goat. This study showed taht the level of S100A7 was upregulated in mammary gland in dairy goat with mastitis. Moreover, the S100A7 concentration in milk has a limited relationship with SCC or mastitis, which show the limited minitor for mastitis in dairy goat.
Specific comments:
1. The title should reveal the conclusion.
2. Blood parameters of dairy goat should be reported, such as glucose, NEFA, BHBA.
3. Limitation of the ELISA kit? Was the sample diluted? Inter and intra CV of ELISA kit?
Author Response
Comment: Mastitis is one of the most significant health problems in dairy goat. This study showed that the level of S100A7 was upregulated in mammary gland in dairy goat with mastitis. Moreover, the S100A7 concentration in milk has a limited relationship with SCC or mastitis, which show the limited minitor for mastitis in dairy goat.
RE: Thanks for reviewer’s valuable and thoughtful comments of mastitis, objective and comprehensive summary of the manuscript. In this study, we explored the relationship between mastitis and antimicrobial peptide S100A7 in dairy; the expression of S100A7 was upregulated in mammary gland of dairy goat with mastitis, and S100A7 concentration in milk was also increased in dairy goat with mastitis, while S100A7 concentration in milk has a limited relationship with somatic cell count or mastitis, which showed the limited minitor for mastitis of S100A7 expression in dairy goat, and provided a new sight of S100A7 role in host defense.
Question 1: 1. The title should reveal the conclusion.
RE: Thanks for reviewer’s helpful and constructive suggestions. In this study, somatic cell count (SCC), milk S100A7 concentration, S100A7 immunohistochemistry and expression of udder were measured in dairy goat. Results showed that the level of S100A7 expression was significantly upregulated in mammary gland of mastitis dairy goats than healthy dairy goats; S100A7 was present weakly in the alveolus of healthy goat mammary gland while densely in collapsed alveolus of mastitis goat mammary gland; S100A7 concentration was significantly upregulated in subclinical mastitis goats and had no significant difference in clinical mastitis goats compare to healthy dairy goats; The S100A7 concentration in milk has a limited relationship with SCC or mastitis.
In a word, although use the conclusion of “the limited minitor for mastitis in dairy goat” is more accurate and capture the key information, the expression of S100A7 in mammary gland will be ignored, so we think the existing title is a comprehensive summary of the full text unless there’s a better choice.
Question 2: 2. Blood parameters of dairy goat should be reported, such as glucose, NEFA, BHBA.
RE: Thanks for reviewer’s helpful and constructive suggestions. In this study, we didn’t detect the relative blood parameters of dairy goat, we just studied explore the relationship between mastitis and S100A7 expression from milk and mammary gland in dairy goat. Of course, the relative blood parameters of are very worthwhile for the detection of blood parameters, which systematically understand the potential role of S100A7 in host defense.
Question 3: 3. Limitation of the ELISA kit? Was the sample diluted? Inter and intra CV of ELISA kit?.
RE: Thanks for reviewer’s reminding. (1) The limits of quantification of Goat S100A7 ELISA kit we used is 1.25 μg/mL - 40 μg/mL; (2) the intra- and inter-assay CV’s is less than 10% and 15% respectively; (3) S100A7 concentration of milk was detected directly with the kit. All the points were revised in the manuscript,
Reviewer 4 Report
Comments and Suggestions for Authors
Ms. Ref. No.: vetsci-2653959R1
Title: “The relationship between mastitis and antimicrobial peptide 2 S100A7 expression in dairy goats”
Veterinary Science
General comments
I have had the opportunity to review the manuscript.
The article is in line with the journal's topic and can help the scientific community better understand the development dynamics of specific diseases in dairy goats. However, there are some parts that need to be elaborated on and others that need to be corrected.
Below are my considerations
Introduction:
L 46 The acronym SCC has not been previously defined. Therefore, please provide the full explanation and then insert the acronym
L 55 clarify the initials 'EC'
Materials and Methods
L 88-90 It is necessary to specify the feed ration with its ingredients and analytical composition (crude protein, crude fat, crude fiber, etc.), as some mastitis issues are associated with the type of nutrition and the forage-to-concentrate ratio, as well as the energy level. It should be described for the animals under consideration, the average weight, the number of lactations, and their age
L 89 change food with feed
L 96-96 Provide detailed information regarding the method of tissue sample collection, how they were stored, transported, refrigerated prior to analysis, and be sure to include specific references
L 118-128 add reference
L 137-144 add the statistical model used by specifying the fixed factor and the variable factor of the equation
Results
L 158 For the first time, ROC is mentioned; please clarify its meaning
Discussion
The discussions need to be expanded depending on the results described.
L 246 it is stated that “In a word, S100A7 plays an important role in innate immune defense of goat mam-246 mary gland, yet there are many factors....” what are these factors? And how could they have influenced your results?
Comments on the Quality of English LanguageThe English language needs improvement; it will be necessary to consult a native speaker author
Author Response
Comment: General comments
I have had the opportunity to review the manuscript.
The article is in line with the journal's topic and can help the scientific community better understand the development dynamics of specific diseases in dairy goats. However, there are some parts that need to be elaborated on and others that need to be corrected.
Below are my considerations:
RE: Thanks for reviewer’s positive comments and offered helpful and constructive suggestions. We have seriously thought about the suggestions and provided our responses in a point-by-point manner.
Introduction
Question 1: L 46 The acronym SCC has not been previously defined. Therefore, please provide the full explanation and then insert the acronym
RE: Thanks for reviewer’s kind reminder. SCC refers to somatic cell count, was defined in line 14 of the simple summary.
Question 2: L 55 clarify the initials 'BC'
RE: Thanks for reviewer’s kind reminder. BC refers to bacteriological culture, was defined in line 24 of the abstract.
Materials and Methods
Question 3: L 88-90 It is necessary to specify the feed ration with its ingredients and analytical composition (crude protein, crude fat, crude fiber, etc.), as some mastitis issues are associated with the type of nutrition and the forage-to-concentrate ratio, as well as the energy level. It should be described for the animals under consideration, the average weight, the number of lactations, and their age
RE: Thanks for reviewer’s advice. The same diets were fed during the experiment, and the nutrient requirements of dairy goats could be satisfied. The diets were referred to the Nutrient Requirements of Small Ruminants (National Research Council, 2007), including alfalfa hay, corn, wheat bran, soybean meal, wheat straw, corn silage, corn germ meal, cottonseed meal, and minerals. The diets and water were provided twice a day at 7:30 and 15:30 throughout the entire trial period. The mentions were revised in the manuscript.
Question 4: L 89 change food with feed
RE: Thanks for reviewer’s advice. “food” was revised into “feed”.
Question 5: L 96-96 Provide detailed information regarding the method of tissue sample collection, how they were stored, transported, refrigerated prior to analysis, and be sure to include specific references
RE: Thanks for reviewer’s advice.
- The normal and mastitis-infected tissues were fixed in 4% paraformaldehyde to perform paraffin sections or immediately frozen in liquid nitrogen until RNA extraction.
- Reference 39 was cited in the method of tissue sample collection.
Question 6: L 118-128 add reference
RE: Thanks for reviewer’ advice.
- Reference 44 and 45 was cited in bacteriological culture.
- Reference 46 was cited in the Immunohistochemistry.
Question 7: L 137-144 add the statistical model used by specifying the fixed factor and the variable factor of the equation
RE: Thanks for reviewer’ advice. The description of Statistics was revised into “The paired groups were applied to compare with unpaired Student’s t-tests after confirming the normal distribution, the multiple groups were applied with one-way analysis of variance (ANOVA) followed by Tukey’s multiple comparison test with GraphPad Prism software (version 8.0)”.
Results
Question 8: L 158 For the first time, ROC is mentioned; please clarify its meaning
RE: Thanks for reviewer’s advice. The Receiver Operating Characteristic (ROC) analysis was performed to compare the sensitivity and specificity of SCC and S100A7, which was revised in the Section 3.2 of the manuscript.
Discussion
Question 9: The discussions need to be expanded depending on the results described.
RE: Thanks for reviewer’s advice, we expanded the discussion depending on the results.
- “Mastitis causes huge economic losses in the milk industry all over the world [3, 6]. So far, antibiotic is the popular method of prevention and treatment for mastitis, but affects the milk quality and causes resistance, so an alternative strategy to prevent and treat mastitis is necessary for the dairy industry development. The AMPs are getting more and more attention for the broad-spectrum antibacterial activity and almost no resistance, which can satisfy the requirement of antibiotic-free milk for humans”
- “In the present study, although bacteriological culture of goat milk was positive in subclinical or clinical mastitis goat, SCC might lower than 1,000 ×103cells/mL, even 500×103 cells/mL; even though the parity of goat in each group was same, there was some degree of fluctuation in the SCC, which further supported that SCC was considered an indirect index in goat”.
- “Various antimicrobial components are synthesized and secreted into milk in the mammary glands, AMPs are the important members. AMPs are potential protein markers of mastitis monitoring, and developed ELISA to monitor mastitis. In the study, the S100A7 concentration not only has a little correlation with SCC, but also the AUC was lower than log10 SCC in subclinical or clinical mastitis goat, and the correlation coefficient between SCC and S100A47 concentration was lower that other AMPs”.
- “Because S100A7 is a calcium-binding protein, so S100A7 concentration is expectedly related to calcium concentration. Circulating leukocytes may be the source of S100A7 for that S100A8-positive cells were leukocytes in goat milk [54], yet the relationship is still unclear. ”.
- “Moreover, S100A8, a member of S100 family, S100A8 concentration in milk was higher than uninfected goat only at the dot of 72 h with LPS treatment, and other times have no significance within 144 h, which showed the complicated phenomenon of S100 protein secretion in goat. S100A7 concentration in goat milk was increased significantly after intramammary infusion with LPS or saline for 24 h, while the expression of S100A7 in milk remains unclear with LPS perfusion in goat mammary gland more than 48 h.”.
- “Type of pathogenic microorganism are equal importance, especially E. coli, which was mainly against invasion by S100A7 antibacterial activity, while has low or no ac-tivity against other pathogenic bacteria, such as Staphylococcus aureus; yet the germline of E. coli didn’t play a decisive role in the severity of its induced mastitis, differences in the main strains causing subclinical and clinical mastitis also lead to differences of S100A7 abundance, and the relationship between S100A7 abundance and the different strains remains unclear. The teat epithelium was considered as the main source of S100A7 in bovine and goat , so the source of keratinocytes is also an important role to study S100A7 expression. S100A7 concentration in milk increased upon frequent teat stimula-tion both with or without milk removal. Moreover, inflammatory cytokines influence its synthesis and secretion, such as IL-1β, can induce S100A7 production”.
- Others were supplemented in the Discussion. The above points were revised in the manuscript.
Question 10: L 246 it is stated that “In a word, S100A7 plays an important role in innate immune defense of goat mammary gland, yet there are many factors....” what are these factors? And how could they have influenced your results?
RE: Thanks for reviewer’s advice.
- Firstly, S100A7 is mainly secreted from the epithelial cells(1), epithelial cells represent a small portion of cells in goat milk (2), so the number of epithelial cells of somatic cells affect its synthesis and secretion.
- Circulating substances may play vital roles in S100A7 synthesis and secretion, the previous reports had shown that butyrate enhanced the production of S100A7 in mammary epithelial cells, and the infusion of butyrate into mammary glands through the teats enhanced S100A7 concentrations(3).
- Type of pathogenic microorganism are equal importance, especially E. coli, which was mainly against invasion by S100A7 antibacterial activity (4).
- The teat epithelium was considered as the main source of S100A7 in bovine (5)and goat (1), so the source of keratinocytes is also an important role to study S100A7 expression.
- Inflammatory cytokines influence its synthesis and secretion, such as IL-1β, can induce S100A7 production (6).
- Other factors, including teat stimulation both with or without milk removal, Circulating leukocytes and others.
These points were revised in the manuscript.
Comments on the Quality of English Language
Question 11: The English language needs improvement; it will be necessary to consult a native speaker author.
RE: Thank you for your valuable and thoughtful comments. We have carefully checked the English writing, and improved it in the revised manuscript according to the suggestion from the native speaker author.
References
- G. W. Zhang, S. J. Lai, Y. Yoshimura, N. Isobe, Messenger RNA expression and immunolocalization of psoriasin in the goat mammary gland and its milk concentration after an intramammary infusion of lipopolysaccharide. Vet J202, 89-93 (2014).
- M. J. Paape, B. Poutrel, A. Contreras, J. C. Marco, A. V. Capuco, Milk Somatic Cells and Lactation in Small Ruminants. Journal of Dairy Science84, E237-E244 (2001).
- Y. Tsugami, N. Suzuki, T. Nii, N. Isobe, Sodium Acetate and Sodium Butyrate Differentially Upregulate Antimicrobial Component Production in Mammary Glands of Lactating Goats. J Mammary Gland Biol Neoplasia27, 133-144 (2022).
- R. Gläser, J. Harder, H. Lange, J. Bartels, E. Christophers, J. M. Schröder, Antimicrobial psoriasin (S100A7) protects human skin from Escherichia coli infection. Nat Immunol6, 57-64 (2005).
- J. Tetens, J. J. Friedrich, A. Hartmann, M. Schwerin, E. Kalm, G. Thaller, The spatial expression pattern of antimicrobial peptides across the healthy bovine udder. J Dairy Sci93, 775-783 (2010).
- D. EisenbeiB, S. Ardebilli, J. Harder, H. Lange, B. Rudolph, J. Schröder, M. Weichenthal, R. Gläser, The antimicrobial protein psoriasin (S100A7) is significantly upregulated in the peripheral blood of psoriasis patients. Journal of Investigative Dermatology126, 61-61 (2006).
Round 2
Reviewer 2 Report
Comments and Suggestions for Authors
Dear authors, thank you for the revision of your manuscript and taking into account my suggestions.
Author Response
Comments: Dear authors, thank you for the revision of your manuscript and taking into account my suggestions.
RE: We would like to thank the reviewer for their accurate and detailed revision of our manuscript. Reviewer offered helpful and constructive suggestions and as a result, we feel that the manuscript has been significantly improved.
Reviewer 4 Report
Comments and Suggestions for Authors
I still remain of the opinion that it is necessary to show not only the ingredients, but also the percentage and analytical composition (crude protein, crude fibre, etc.) see L 85-88 and add this information before acceptance.
Comments on the Quality of English LanguageOk
Author Response
Comments and Suggestions for Authors:I still remain of the opinion that it is necessary to show not only the ingredients, but also the percentage and analytical composition (crude protein, crude fibre, etc.) see L 85-88 and add this information before acceptance.
RE: Thanks for reviewer’s helpful and constructive suggestions. We have seriously thought about the suggestions, and added the ingredients and chemical composition of the postpartum diets as a supplemental table 1.
Supplemental Table 1. Ingredients and chemical composition of the postpartum diets on the dry matter (DM) basis
Ingredient (% of DM) |
Postpartum |
Nutrient composition |
Postpartum |
Alfalfa hay |
18.42 |
DM (% of fresh) |
48.20 |
Corn |
23.16 |
Neutral detergent fiber (NDF, %) |
37.70 |
Wheat bran |
8.37 |
Acid detergent fiber (ADF, %) |
17.40 |
Soybean meal |
9.02 |
Crude protein (CP, %) |
16.40 |
Wheat straw |
0.00 |
Starch (%) |
25.40 |
Corn Silage |
30.66 |
Ether extract (%) |
3.20 |
Corn germ meal |
3.26 |
Calcium (Ca, %) |
0.66 |
Cottonseed meal |
5.12 |
Phosphorus (P, %) |
0.37 |
Calcium hydrophosphate |
0.51 |
Magnesium (Mg, %) |
0.19 |
Limestone |
0.46 |
Sulfur (S, %) |
0.20 |
Sodium carbonate |
0.38 |
Chloride (Cl, %) |
0.25 |
Sodium chloride |
0.45 |
DCAD (mEq/kg of DM)2 |
+733 |
Mineral and vitamin premix 1 |
0.19 |
NEL (Mcal/kg)3 |
1.62 |
1 The mineral-vitamin premix provided the following per kg of diets: vitamin A 250,000 IU, vitamin D 23,250 IU, vitamin E 1500 IU, manganese 800 mg, zinc 1800 mg, copper 370 mg, iron 2200 mg, cobalt 50 mg, iodine 30 mg, selenium 30 mg.
2 The dietary cation-anion difference (DCAD) was calculated using the formula DCAD (mEq/kg of DM) = (Na + K) − (Cl++− + S2−).
3 Net energy for lactation (NEL) was estimated using CPM-Dairy software (version 3.0.8.1).
Comments on the Quality of English Language: Ok
RE: Thanks you very much for the reviewer’s previous suggestions of English language improvement, we have carefully checked the English writing, and improved it in the revised manuscript.

Round 3
Reviewer 4 Report
Comments and Suggestions for Authors
Ok
Comments on the Quality of English LanguageOk